environmental chemistry

nitrification inhibitor, nitrapyrin, adsorption, black soil, chernozem, planosol

**Author for correspondence:**
Qiang Gao
e-mail: jlauzy2019@163.com

This article has been edited by the Royal Society of Chemistry, including the commissioning, peer review process and editorial aspects up to the point of acceptance.

# The adsorption and mechanism of the nitrification inhibitor nitrapyrin in different types of soils

Zhongqing Zhang[1], Qiang Gao[1], Jingmin Yang[1], Yue Li[2], Jinhua Liu[1], Yujun Wang[1], Hongge Su[1], Yin Wang[1], Shaojie Wang[1] and Guozhong Feng[1]

[1]Key Laboratory of Soil Resource Sustainable Utilization for Jilin Province Commodity Grain Bases, College of Resources and Environmental Science, Jilin Agricultural University, Changchun 130118, People's Republic of China
[2]Ecological Environment Bureau of the Changchun Jingyue Economic Development Zone, Changchun 130118, People's Republic of China

ZZ, 0000-0003-4866-3340; QG, 0000-0002-5815-3048

The nitrapyrin was easily adsorbed by soil, but most current studies have focused on comparing the effects of nitrapyrin application at different soil organic matter levels and in different soil types. The adsorption kinetics and isotherm adsorption of the nitrification inhibitor nitrapyrin in black soil, chernozem and planosol were studied in this paper. The adsorption kinetics were fitted by quasi-second-order kinetic equation ($R^2 \geq 0.8907$, $p < 0.05$) with a lower acting energy of adsorption ($E_a < 8.0 \, \text{kJ mol}^{-1}$). The isotherm was fitted by the Langmuir equation ($R^2 \geq 0.9400^*$, $p < 0.05$). The adsorption mechanism was determined to involve a spontaneous endothermic reaction accompanied mainly by physical adsorption to the surface that belonged to the 'L' isotherm curve ($n > 1$). Temperature promoted the adsorption of nitrapyrin in these three soils, and the maximum adsorption occurring at different temperatures following the order of black soil > planosol > chernozem. The adsorption capacity and rate decreased with decreasing soil organic matter. For the black soil, the nitrapyrin EC adsorption rate was more than seven times higher than that of nitrapyrin CS. The result would determine the dose of nitrapyrin required for availability in different types of soils and to provide a theoretical basis for elucidating the adsorption of nitrapyrin in the soil environment.

## ROYAL SOCIETY OF CHEMISTRY

# 1. Introduction

The application of nitrification inhibitors can effectively improve nitrogen use efficiency, increase crop yield and quality and reduce environmental pollution caused by the excessive use of nitrogen fertilizer. Thus, the application of nitrification inhibitors provides a promising method for reducing the amount of nitrogen fertilizer application, which was beneficial to farmers' incomes and environmental protection. Nitrapyrin was an early-developed nitrification inhibitor with a remarkable nitrification inhibiting effect. Goring [1] first reported that nitrapyrin had nitrification inhibiting properties. In 1974, the DOW Chemical Company developed the nitration inhibition product 'N-server' using nitrapyrin. At present, the product has been widely used in agricultural production, and the research on nitrapyrin focuses mainly on the participation of the nitrogen cycle and its impact on crop yields, such as significantly inhibiting the activity of nitrifying bacteria [2,3], delaying the transformation of ammonium nitrogen to nitrate nitrogen [4,5], and reducing nitrogen leaching and NO emissions [6–8], which improve nitrogen use efficiency and in turn increase crop yield [9–11].

The availability of nitrification inhibitors was affected by a variety of external factors, such as soil clay composition, soil organic matter, pH, temperature, soil water content and soil microorganisms [12–15]. In particular, soil organic matter was one of the important factors governing the effectiveness of nitrification inhibitors because of adsorption [14,16–20]. Previous studies have shown that nitrapyrin, when applied to soil, was easily adsorbed by soil organic matter [21], especially for soils with high organic matter, and its availability was significantly reduced [22–24]. However, most current studies were focused on comparing the effects of nitrapyrin application at different soil organic matter levels and in different soil types. Gu *et al.* [25] analysed the nitrification inhibition effect of nitrapyrin on sandy and clayey soil with different organic matter contents, and the overall performance in sandy soil was greater than that in clayey soil.

Because the availability and environmental fate of the chemicals applied to the soil were controlled by the adsorption process, which were determined by the physical and chemical properties of the soil [26,27] such as the content and composition of soil organic matter, it was deduced that the availability of nitrification inhibitors applied to soil would also be controlled by the soil adsorption process. Some scholars have carried out relevant research in this field. For example, Jacinthe & Pichtel [28] have studied the adsorption mechanisms of nitrapyrin and dicyandiamide onto humus, pointing out that adsorption mainly occurs through the ionic bonding mechanism. Biologically, the ionic bonding mechanism involves the amino group of dicyandiamide or the N of the nitrapyrin ring and the functional groups of the negatively charged humus. Rawat *et al.* [29] studied the kinetics and adsorption–desorption behaviour of nitrification inhibitor AM in soils. The adsorption of AM onto soil samples obtained at different depths followed pseudo-second-order reaction kinetics, and the adsorption isotherm was of the S-type, which was determined by the Freundlich isotherm model. However, there were few reports on the adsorption kinetics and isotherm adsorption of nitrapyrin in different types of soils.

Northeast China has a 'Golden Corn Belt', which was also one of the three major black soil regions in the world, with a high content of soil organic matter. Local farmers typically use one fertilizer application, which results in low nitrogen use efficiency [30,31]. Under the nitrogen fertilizer reduction policy formulated by the Chinese government, adding a certain amount of nitrification inhibitor to chemical fertilizer will provide an effective way to improve nitrogen use efficiency and reduce nitrogen fertilizer application. However, adsorption by organic matter must also be considered after nitrapyrin was applied to black soil because it has a relatively high organic matter content.

In this study, three types of soils (black soil, chernozem and planosol) in northeast China and two dosage forms of nitrapyrin were selected. The adsorption kinetics and isotherm adsorption of nitrapyrin in three types of soils were analysed with the methods of classical batch equilibrium. The present work was an attempt to assess the adsorption mechanism on the adsorption behaviour of nitrapyrin on three soils in China, which will help to determine the dose of nitrapyrin required for the availability of this nitrification inhibitor in different types of soils and to provide a theoretical basis for elucidating the adsorption of nitrapyrin in the soil environment. At the same time, this study provides basic theoretical support for developing new dosage forms of nitrapyrin to reduce the adsorption of organic matter and improve the effectiveness of nitrapyrin application.

# 2. Material and methods

## 2.1. Test soil

Three kinds of soils, including black soil, chernozem and planosol, were selected. The black soil was located in Hailun City, Heilongjiang Province (E126°50′55″, N47°26′31″), the chernozem and the planosol were

**Table 1.** Basic physical and chemical properties of the study soils.

| soil type | pH | organic matter (g kg$^{-1}$) | N (%) | P$_2$O$_5$ (%) | K$_2$O (%) |
|---|---|---|---|---|---|
| black soil | 6.55 | 48.51 | 1.35 | 0.32 | 0.15 |
| chernozem | 7.92 | 32.66 | 1.01 | 0.27 | 0.11 |
| Planosol | 4.92 | 24.92 | 0.35 | 0.12 | 0.07 |

**Table 2.** Treatments and organic matter contents.

| treatment | H$_2$O$_2$ ratio (%) | organic matter (g kg$^{-1}$) |
|---|---|---|
| treatment 1 | 0 | 48.51 |
| treatment 2 | 30 | 44.30 |
| treatment 3 | 60 | 34.37 |
| treatment 4 | 100 | 22.35 |
| treatment 5 | — | 7.27 |

located in Nongan City, Jilin Province (E125°07′37″, N44°17′22″) and Yongji City, Jilin Province (E125°48′43″, N43°25′33″), respectively. The land was planted with corn for many years without application of the nitrification inhibitor nitrapyrin. The soil was sampled at a depth of 0–20 cm, and the surface plant residues were removed. After air drying, the soil samples were passed through a 100 mesh nylon screen and stored in sealed plastic bags. The basic physical and chemical properties of the three soils are shown in table 1.

For removal of organic matter from the soil, 200 g of black soil was added to a 500 ml beaker, and the soil was moisturized by the addition of a small amount of distilled water. Then, a corresponding amount of 30% H$_2$O$_2$ solution was added according to table 2. After thorough mixing, the surface was covered with a dish, and the soil was heated in a water bath at 80 ± 5°C to oxidize it. After the soil was dried in an oven at 105°C, it was ground and passed through a 100 mesh nylon screen. The soil organic matter content was determined by the potassium dichromate volumetric method. For treatment 5, when the foam settled, more H$_2$O$_2$ was added until the soil colour became light, which indicated that there was no reaction. The soil organic matter was tested through reference to the book of 'Technical guidance for soil agrochemical analysis'.

## 2.2. Chemicals

Ethanol (analytical grade) was purchased from Beijing Chemical Co., Ltd, and acetonitrile (chromatography grade) was purchased from Shandong Yuwang and World New Materials Co., Ltd. Nitrapyrin raw medicine (98.0%) was purchased from Wuhan Yuancheng Gongchuang Technology Co., Ltd, and nitrapyrin standard (99.0%) was purchased from LGC Labort GmbH·Bgm, Germany. Calcium chloride (analytical grade) was obtained from Huoguo Pharmaceutical Group Co., Ltd. The water used in the experiment was twice deionized water.

## 2.3. Instruments

High-performance liquid chromatographer (Agilent 1260, USA): six-way valve injector, ultraviolet detector, Agilent SB-C18 stainless steel chromatographic column (4.6 × 150 mm, 5 µm); high-speed refrigerated centrifuge (Z36HK, HERMLE); constant temperature oscillating water bath (SHA-C, Huapuda Instrument Co., Ltd).

## 2.4. Adsorption kinetics

First, 1.0000 g soil was weighed accurately and placed into a 50 ml plug centrifuge tube, and a 25 ml nitrapyrin solution of 50 mg l$^{-1}$ (0.01 mol l$^{-1}$ CaCl$_2$ ethanol/water (v : v = 5 : 95) solution) was added,

followed by oscillation at 200 r min$^{-1}$ for 5, 10, 30, 60, 120, 240, 360, 480, 720 and 1440 min at 298, 308 and 318 K in dark conditions. The solution was centrifuged at a rate of 10 000 r min$^{-1}$, and 5 ml supernatant was filtered through a 0.45 µm organic filter membrane into a brown liquid spectrum bottle. Then, the bottle was sealed and stored at 3°C in the dark. The content of nitrapyrin was determined by high-performance liquid chromatography (HPLC). Three repetitions were performed for each treatment.

## 2.5. Adsorption isotherm and thermodynamics

The three kinds of soil were accurately weighed to 1.0000 g and placed in a 50 ml plug centrifuge tube, and 25 ml nitrapyrin solution (0.01 mol l$^{-1}$ CaCl$_2$ ethanol/water (v : v = 5 : 95) solution) was added to the tube. The concentrations of the nitrapyrin solutions were 0, 5, 10, 15, 25, 50 and 75 mg l$^{-1}$ for a total of seven treatments that were repeated three times. Light was avoided, and the tubes were placed in a constant temperature oscillating water bath (200 r min$^{-1}$). The adsorption experiments were carried out at different reaction temperatures (298, 308, 318 K). After 24 h, the supernatant was centrifuged at a rate of 10 000 r min$^{-1}$, and 5 ml supernatant was filtered out by a 0.45 µm organic filter membrane and sealed in a brown liquid spectrum bottle, which was stored in the dark at 3°C. The content of nitrapyrin was determined by HPLC.

## 2.6. Effects of different factors on adsorption

### 2.6.1. Soil organic matter content

One gram of soil from which the organic matter had been removed was accurately weighed and placed in a 50 ml plug centrifuge tube, and 25 ml nitrapyrin solution (0.01 mol l$^{-1}$ CaCl$_2$ ethanol/water (v : v = 5 : 95) solution) was added. The concentrations of nitrapyrin solution were 0, 5, 10, 15, 25, 50 and 75 mg l$^{-1}$ for a total of seven treatments, with three repetitions. Light was avoided, and samples were placed in a constant temperature oscillating water bath (200 r min$^{-1}$). The adsorption test was carried out at 298 K. After 24 h, the supernatant was centrifuged at a rate of 10 000 r min$^{-1}$, and 5 ml supernatant was filtered through a 0.45 µm organic filter membrane and sealed in a brown liquid spectrum bottle, which was stored in the dark at 3°C. The content of nitrapyrin was determined by HPLC.

### 2.6.2. Nitrapyrin in different dosage forms

Under the optimum instrumental conditions, black soil was selected as the test soil, and two dosage forms of nitrapyrin (EC and CS) were selected. One gram of black soil was accurately weighed and placed in a 50 ml plug centrifuge tube, and 25 ml nitrapyrin solution (0.01 mol l$^{-1}$ CaCl$_2$ ethanol/water (v : v = 5 : 95) solution) was added. The concentrations of nitrapyrin solution were 0, 2, 4, 6, 10, 15, 25, 50 and 75 mg l$^{-1}$ for a total of nine treatments, which were repeated three times. Light was avoided, and samples were placed in a constant temperature oscillating water bath (200 r min$^{-1}$). The adsorption test was carried out at 298 K. After 24 h, the supernatant was centrifuged at a rate of 10 000 r min$^{-1}$, and 5 ml supernatant was filtered through by a 0.45 µm organic filter membrane and sealed in a brown liquid spectrum bottle, which was stored in the dark at 3°C. The content of nitrapyrin was determined by HPLC.

## 2.7. High-performance liquid chromatography conditions

The HPLC conditions were as follows: the detector, UV–Vis (Agilent 1260, USA); the wavelength, 285 µm; the chromatographic column, C18 column (4.6 × 150 mm, 5 µm, Agilent Pursuit); the column temperature, 25°C; the mobile phase, acetonitrile/water (8/2); the flow rate, 0.8 ml min$^{-1}$; the injection volume, 20 µl. The concentration gradient of nitrapyrin was 2, 4, 8, 10, 15, 25, 50 and 75 mg l$^{-1}$.

## 2.8. Statistical analysis

Origin 8.5 software was used to fit the data. Agilent workstation (ChemStation Edition for LC and LC/MS systems, C0.01.05 [35]) was used to obtain the chromatogram of nitrapyrin. The correlation analysis model and calculation formula were as follows.

(1) Adsorption capacity and adsorption rate

$$Q_e = \frac{(C_0 - C_e)V}{m}$$ (2.1)

and

$$W_e = (C_0 - C_e)V \times 100/(C_0 V).$$ (2.2)

'$Q_e$' was the adsorption capacity (mg kg$^{-1}$). '$W_e$' was the adsorption rate (%). '$C_0$' was the initial concentration of the nitrapyrin solution (mg l$^{-1}$). '$C_e$' was the adsorption equilibrium solution concentration (mg l$^{-1}$). '$V$' was the adsorption solution volume (ml). '$m$' was the weight of the soil (g).

(2) Dynamic model

Quasi-first-order dynamic equation

$$\ln(q_e - q_t) = \ln q_{e1} - k_1 t.$$ (2.3)

Quasi-second-order kinetic equation

$$\frac{t}{q_t} = \frac{1}{k_2 q_e^2} + \frac{t}{q_e}.$$ (2.4)

Elovich equation

$$q_t = a + b_t.$$ (2.5)

Intraparticle diffusion equation

$$q_t = k_p t^{0.5}.$$ (2.6)

'$q_t$' was the adsorption amount when time was $t$ (mg kg$^{-1}$). '$t$' was the adsorption time (min). '$a$, $k$' was a model parameter. '$q_e$' was the maximum adsorption capacity (mg kg$^{-1}$). '$A$' was a constant related to the initial rate of reaction. '$b$' was a constant related to the activation energy of adsorption.

(3) Activation energy parameter

$$\ln k_2 = \ln A - \frac{E_a}{RT},$$ (2.7)

$$\ln\left(\frac{k_2}{T}\right) = \left[\ln\left(\frac{kB}{h}\right) + \left(\frac{\Delta S^{\#}}{R}\right)\right] - \left(\frac{\Delta H^{\#}}{RT}\right)$$ (2.8)

and

$$\Delta G^{\#} = \Delta H^{\#} - T\Delta S^{\#}.$$ (2.9)

'$E_a$' was the adsorption activation energy. '$\Delta G^{\#}$' was the activation free energy. '$\Delta H^{\#}$' was the activation enthalpy. '$\Delta S^{\#}$' was the activation entropy [32]. '$K_2$' was the rate constant of the second-order adsorption kinetic equation and '$T$' was the reaction temperature. '$A$', '$R$', '$kB$' and '$h$' were the Arrhenius constant, universal gas constant (8.314 J (K mol$^{-1}$)$^{-1}$), Boltzman constant (1.3807 × 10$^{-23}$ J K$^{-1}$) and Planck constant (6.6261 × 10$^{-34}$ J s), respectively. According to the slopes of $\ln k_2$ and $1/T$ linear graphs, the $E_a$ values can be obtained. The $\Delta H^{\#}$ and $\Delta S^{\#}$ values can be obtained according to the slopes and interceptions of $\ln(K_2/T)$ and $1/T$ linear graphs.

(4) Adsorption isotherm equation

Langmuir equation

$$Q_d = \frac{k_l Q_{d0} C_d}{1 + k_l C_d}.$$ (2.10)

Freundlich equation

$$Q_d = k_f C_d^{1/n}.$$ (2.11)

'$Q_d$' represents the adsorption amount (mg kg$^{-1}$). '$C_d$' was the adsorption equilibrium solution concentration (mg l$^{-1}$). '$Q_{d0}$' was the maximum adsorption capacity (mg kg$^{-1}$). '$k$' was a constant.

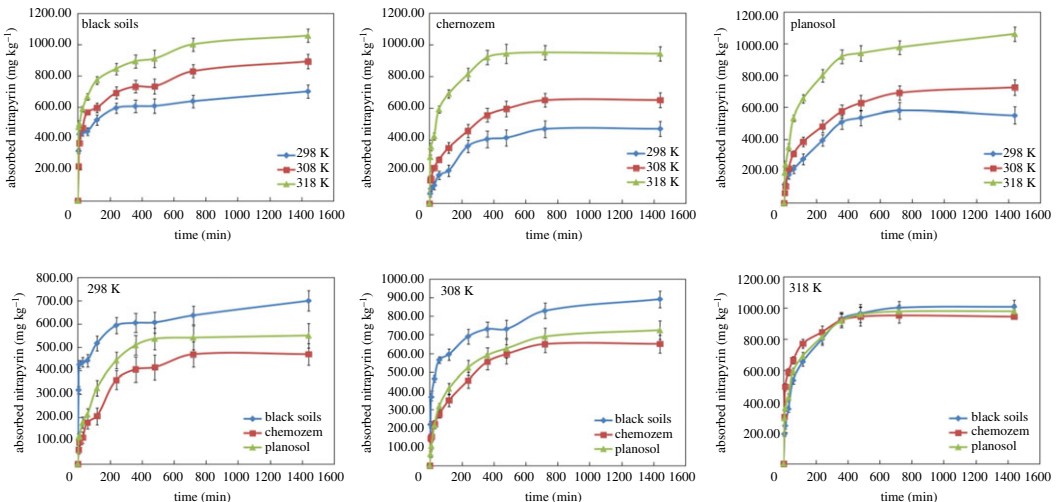

**Figure 1.** Adsorption kinetics curve (the initial concentration was 50 mg l$^{-1}$).

(5) Thermodynamic adsorption parameters

$$\Delta G^0 = -\mathrm{RT}\ \ln k_l, \tag{2.12}$$

$$\Delta H^0 = R\left[\frac{T_2 T_1}{T_2 - T_1}\right]\ln\left(\frac{k_{l(T2)}}{k_{l(T1)}}\right) \tag{2.13}$$

$$\text{and}\quad \Delta S^0 = \frac{\Delta H^0 - \Delta G^0}{T}. \tag{2.14}$$

'$\Delta G^0$' was the free energy change (kJ mol$^{-1}$). '$\Delta H^0$' was the enthalpy change (kJ mol$^{-1}$). '$\Delta S^0$' was the entropy change (J (mol K$^{-1}$)$^{-1}$), '$R$' was the gas constant (8.314 J (K mol$^{-1}$)$^{-1}$). '$T$' was the absolute temperature (K). '$k_l$' was the equilibrium constant (l mol$^{-1}$) in the Langmuir equation.

(6) Adsorption mechanism

$$Q_T = Q_P + Q_A, \tag{2.15}$$

$$Q_P = K_{oc}f_{oc}C_e \tag{2.16}$$

$$\text{and}\quad Q_A = KC_e^n - K_{oc}f_{oc}C_e, \tag{2.17}$$

where $Q_T$ was the total adsorption amount of nitrapyrin (mg kg$^{-1}$), $Q_P$ was the adsorption amount of nitrapyrin due to distribution in the process of adsorption (mg kg$^{-1}$), $Q_A$ was the adsorption amount of nitrapyrin due to surface adsorption in the process of adsorption (mg kg$^{-1}$), $K_{oc}$ was the distribution coefficient of organic carbon standardized according to the $K_{oc}$ equation of the $K_d$ value (l kg$^{-1}$), and $f_{oc}$ was the organic carbon content in the tested soil [33,34].

# 3. Results and discussion

## 3.1. Adsorption kinetics

The adsorption of organic matter in soil was generally divided into two stages: rapid adsorption occurs in the first stage and slow adsorption occurs in the second stage [35]. Figure 1 shows the adsorption kinetics curves of nitrapyrin in three soil types with initially rapid adsorption stage and following slow adsorption stage to the equilibrium after 400 min. Because nitrapyrin molecules occupied the easily adsorbed hydrophobic sites on the soil organic matter surface at the initial stage, the adsorption capacity increased rapidly. In the later stage, when most hydrophobic sites were occupied by nitrapyrin, adsorption mainly occurred in the pore structure of soil organic matter, resulting in a decrease in the adsorption rate and capacity. At different temperatures, the overall amount of nitrapyrin adsorption in the three types of soils showed the trend of black soil > planosol > chernozem.

The kinetics data of nitrapyrin adsorption in three kinds of soils were fitted by a quasi-first-order kinetic equation, quasi-second-order kinetic equation, Elovich equation and intraparticle diffusion equation. It can be seen from table 3 that the coefficients of determination showed the following

**Table 3.** Kinetics equations.

| soil | temperature $T$ (K) | quasi-first-order dynamic equation | | | quasi-second-order kinetic equation | | | Elovich equation | | | intraparticle diffusion equation | |
|---|---|---|---|---|---|---|---|---|---|---|---|---|
| | | $q_e$ (mg kg$^{-1}$) | $k_1$ | $R^2$ | $q_e$ (mg kg$^{-1}$) | $k_2$ | $R^2$ | $a$ | $b$ | $R^2$ | $k$ | $R^2$ |
| black soil | 298 | 527.01 | 99.42 | 0.6265 | 601.25 | 0.000289 | 0.8991 | −120.91 | 83.57 | 0.9409 | 26.25 | −0.3599 |
| | 308 | 607.92 | 114.32 | 0.3996 | 783.78 | 0.000174 | 0.9397 | 80.04 | 110.93 | 0.9919 | 32.21 | 0.3212 |
| | 318 | 777.55 | 145.23 | 0.3903 | 1004.01 | 0.000116 | 0.8907 | 94.84 | 143.46 | 0.9848 | 41.60 | 0.4101 |
| chernozem | 298 | 276.80 | 52.04 | 0.1409 | 524.13 | 0.000038 | 0.9693 | −120.91 | 83.57 | 0.9409 | 16.56 | 0.8269 |
| | 308 | 405.74 | 76.50 | 0.2128 | 679.05 | 0.000018 | 0.9411 | −89.91 | 104.16 | 0.9563 | 23.31 | 0.7526 |
| | 318 | 741.35 | 138.69 | 0.4721 | 918.56 | 0.000016 | 0.9428 | 185.19 | 116.87 | 0.9770 | 37.93 | −0.0117 |
| planosol | 298 | 352.51 | 66.22 | 0.1920 | 597.83 | 0.000021 | 0.9634 | 352.51 | 66.22 | 0.9920 | 20.31 | 0.7236 |
| | 308 | 426.93 | 80.23 | 0.1554 | 751.84 | 0.000014 | 0.9850 | −182.50 | 128.07 | 0.9906 | 25.30 | 0.8112 |
| | 318 | 699.37 | 130.81 | 0.3403 | 969.34 | 0.000010 | 0.9772 | 29.24 | 140.82 | 0.9777 | 37.64 | 0.4238 |

**Table 4.** Kinetic activation parameters.

| soil | temperature $T$ (K) | kinetic activation parameters | | | |
| | | $E_a$ (kJ mol$^{-1}$) | $\Delta G^{\#}$ (kJ mol$^{-1}$) | $\Delta H^{\#}$ (kJ mol) | $\Delta S^{\#}$ (J mol$^{-1}$ K$^{-1}$) |
|---|---|---|---|---|---|
| black soil | 298 | 7.5 | 85.2 | 59.5 | −86.4 |
| | 308 | | 86.1 | | |
| | 318 | | 86.9 | | |
| chernozem | 298 | 7.1 | 92.9 | 56.7 | −121.5 |
| | 308 | | 94.2 | | |
| | 318 | | 95.4 | | |
| planosol | 298 | 4.6 | 78.7 | 38.4 | **−135.4** |
| | 308 | | 80.1 | | |
| | 318 | | 81.4 | | |

trend: Elovich equation > quasi-second-order kinetic equation > quasi-first-order kinetic equation > intraparticle diffusion equation, in which the coefficients of determination of the Elovich equation and the quasi-second-order kinetic equation were significant. However, the constant related to the initial rate of reaction in the parameters of the Elovich equation was negative, so the quasi-second-order kinetic equation was used to fit the nitrapyrin kinetics of the three kinds of soil adsorption ($R^2 \geq$ 0.8907, $p < 0.05$). With the increase in reaction temperature, the adsorption rate constant ($K_2$) of the quasi-second-order kinetic equation decreased, and the increase in temperature prolonged the time for nitrapyrin to reach adsorption equilibrium. The amount of adsorption in different types of soils was different when adsorption equilibrium was reached. The maximum equilibrium adsorption capacity at different temperatures showed the following trend: black soil > planosol > chernozem.

## 3.2. Kinetic activation parameters

According to the linear relationship between $\ln k_2 - 1/T$ and $\ln (K_2/T) - 1/T$, the adsorption activation parameters are shown in table 4. The nitrapyrin adsorption activation energy ($E_a$) in three soils was black soil > chernozem > planosol with values less than 8.0 kJ mol$^{-1}$, which shows that the amount of energy required was small and that the physical adsorption process played the main role. Among the thermodynamic activation parameters, all three types of soils needed energy to adsorb nitrapyrin because $\Delta G^{\#}$ was larger than 0. Furthermore, increasing temperature promoted $\Delta G^{\#}$, which was beneficial for the adsorption of nitrapyrin. $\Delta H^{\#} > 0$ and $\Delta S^{\#} < 0$, indicating that the adsorption of nitrapyrin in soils includes an endothermic reaction and that its activation energy decreases. The relatively lower values of $E_a$, $\Delta H^{\#}$ and $\Delta G^{\#}$ and higher values of $\Delta S^{\#}$ for chernozem indicated that the adsorption energy, activation entropy and adsorption rate of nitrapyrin in this soil were lower and the adsorption rate was slower compared with those of the other soils.

## 3.3. Adsorption isotherm

The amount of nitrapyrin adsorption in the three types of soils increased with increasing nitrapyrin concentration. The amount of nitrapyrin adsorption also increased with increasing temperature (figure 2).

The Freundlich model and Langmuir model were often used to fit adsorption kinetics [36,37]. The Freundlich model was mainly suitable for adsorbents with non-uniform surfaces. It was considered that the adsorption between molecules and adsorbents was via multilayer inhomogeneous adsorption [38–40]. When the adsorption kinetics parameter '$n$' was more than 1, the adsorbed compound belonged to the L-type adsorption isotherm curve of surface monolayer adsorption. The Langmuir model was mainly used to describe the monolayer surface adsorption process. It was considered that the adsorption energy of each molecule was the same and had nothing to do with the degree of coverage on the surface of the adsorbents. Moreover, the adsorption of organic compounds only occured in the fixed position of the adsorbents, and there was no interaction between the adsorbents.

The adsorption isotherms of nitrapyrin in the three types of soils were fitted by Langmuir and Freundlich equations. Table 5 shows that both the Langmuir and Freundlich equations can be used to fit

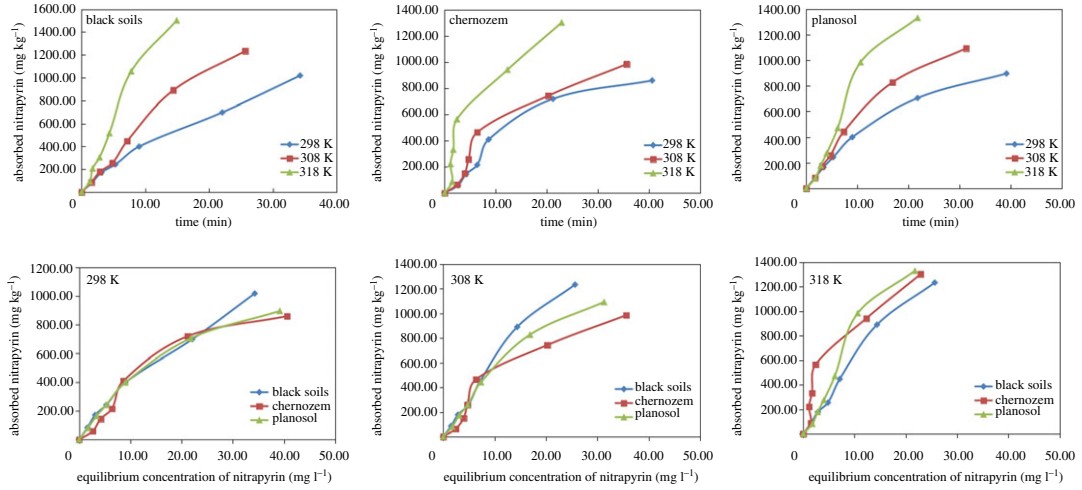

**Figure 2.** Adsorption isotherm curves.

**Table 5.** Isotherm equations.

| soil type | temperature $T$ (K) | Langmuir model | | | Freundlich model | | |
|---|---|---|---|---|---|---|---|
| | | $q_m$ | $k_L$ | $R^2$ | $k_F$ | $n$ | $R^2$ |
| | | (mg kg$^{-1}$) | (l mg$^{-1}$) | _ | (mg kg$^{-1}$ (mg l$^{-1}$)$^{-1/n}$) | _ | _ |
| black soil | 298 | 2372.28 | 4847.22 | 0.9931 | 759 419.84 | 1.3300 | 0.9969 |
| | 308 | 4053.00 | 4057.71 | 0.9900 | 2 722 130.00 | 1.1900 | 0.9826 |
| | 318 | 5859.35 | 5521.55 | 0.9782 | 7 685 840.00 | 1.1400 | 0.9700 |
| chernozem | 298 | 1558.45 | 7735.41 | 0.9600 | 305 357.40 | 1.4910 | 0.9300 |
| | 308 | 1685.28 | 9193.50 | 0.9600 | 358 415.33 | 1.4976 | 0.9400 |
| | 318 | 1691.18 | 29068.31 | 0.9400 | 223 211.09 | 1.7963 | 0.9300 |
| planosol | 298 | 1498.73 | 9007.26 | 0.9897 | 116 873.62 | 1.7575 | 0.9540 |
| | 308 | 2234.31 | 7349.96 | 0.9923 | 704 001.73 | 1.3867 | 0.9758 |
| | 318 | 4182.20 | 5204.28 | 0.9584 | 3 545 350.00 | 1.1833 | 0.9437 |

the adsorption isotherms of nitrapyrin in the three soils ($R^2 > 0.9300$, $p < 0.05$), with the Langmuir equation ($R^2 > 0.9400$, $p < 0.05$) being better than the Freundlich equation ($R^2 > 0.9300$, $p < 0.05$). The fitting effect of the Langmuir equation in black soil was better than that in the chernozem and planosol. Therefore, the Langmuir equation was adopted to fit the adsorption isotherms of nitrapyrin in the three kinds of soils.

According to the Langmuir equation, the maximum amount of nitrapyrin adsorption ($q_m$) in the three types of soils at different temperatures showed the following trend: black soil > planosol > chernozem. The constant of the Freundlich equation ($n > 1$) indicated that the adsorption of nitrapyrin in the three types of soils occurs relatively easily in spontaneous processes and belongs to the L-type isotherm curve, which indicates that at the beginning of adsorption, nitrapyrin molecules quickly occupy the adsorption sites on the soil surface, mainly via surface adsorption. With the increase in reaction temperature, the Langmuir isotherm parameter ($k_L$) increased, which also indicated that the increase in temperature could promote the adsorption of nitrapyrin in the three types of soils.

## 3.4. Thermodynamic parameters

The change in the free energy ($\Delta G^0$) of organic matter during the process of adsorption by soil can be used as a basis for measuring the adsorption strength and driving force of different soils. When the absolute value of adsorption free energy ($\Delta G^0$) was greater than $40 \text{ kJ mol}^{-1}$, chemical adsorption played an important role; otherwise, physical adsorption is the main process.

**Table 6.** Thermodynamic parameters.

| soil | temperature $T$ (K) | thermodynamic adsorption parameters | | |
|---|---|---|---|---|
| | | $\Delta G^0$ (kJ mol$^{-1}$) | $\Delta H^0$ (kJ mol$^{-1}$) | $\Delta S^0$ (J mol$^{-1}$ K$^{-1}$) |
| black soil | 298 | −36.9 | 5.0 | 144.7 |
| | 308 | −37.7 | | |
| | 318 | −39.8 | | |
| chernozem | 298 | −38.0 | 49.2 | 301.4 |
| | 308 | −39.7 | | |
| | 318 | −44.0 | | |
| planosol | 298 | −39.7 | 21.7 | 60.6 |
| | 308 | −40.5 | | |
| | 318 | −40.9 | | |

The thermodynamic adsorption parameters calculated according to the linear calculation of $\Delta G^0 - T$ are listed in table 6. The $\Delta G^0$ values of the three soils were all negative, and the absolute value of $\Delta G^0$ increased with increasing temperature, which indicates that the increase in temperature was beneficial to the process of chemical adsorption of nitrapyrin in soil. The adsorption of nitrapyrin in the three soils was an easy and spontaneous process, and the adsorption capacity increased with increasing temperature, which was consistent with the trends revealed by the adsorption kinetics and adsorption isotherm parameters mentioned above. The absolute values of $\Delta G^0$ in the three soils were all less than 40 kJ mol$^{-1}$ except in the chernozem (318 K) and planosol (308 and 318 K). This showed that the adsorption of nitrapyrin in three types of soils at 298 K was mainly physical, while part of chemical adsorption occurred in the chernozem and planosol at 308 and 318 K. The increase in temperature was beneficial to the process of chemical adsorption of nitrapyrin in soil.

The adsorption enthalpy change ($\Delta H^0$) was the result of the interaction of various forces between the adsorbents, which reflects the properties of the forces between the adsorbents, and the contributions of different forces to the adsorption enthalpy were different [41]. Because the adsorption process of organic pollutants on the solid–liquid interface was usually the result of a variety of adsorption forces, it was speculated that there were different forces in the adsorption process of nitrapyrin in different types of soils. That is, the possible adsorption mechanism of nitrapyrin in different types of soils was inferred from the adsorption enthalpy change parameters. In this study, we found that there were significant differences in the adsorption enthalpy variation ($\Delta H^0$) of nitrapyrin in three types of soils. The $\Delta H^0$ values in black soil, chernozem and planosol were 5.01 kJ mol$^{-1}$, 49.24 kJ mol$^{-1}$ and 21.68 kJ mol$^{-1}$, respectively. However, the adsorption forces were mainly hydrogen bonds and dipolar forces.

The standard entropy change ($\Delta S^0$) reflects the disorder and degree of freedom of the solid–liquid interface in the adsorption process [42]. In this study, the standard entropy change in nitrapyrin adsorption in three types of soils was greater than zero, which was mainly due to the increase in entropy change in nitrapyrin not adsorbed by soil being greater than the decrease in entropy change in nitrapyrin adsorbed by soil. Therefore, the positive value of $\Delta S^0$ indicates that the nitrapyrin adsorption reaction in the three types of soils was an endothermic process, and the adsorption process contributed to the increase in disorder and degree of freedom of the solid–liquid interface.

## 3.5. Adsorption mechanism

The comprehensive adsorption mechanism can be used to explain nonlinear adsorption phenomena, which include partition adsorption and surface adsorption [43,44]. To explore the adsorption mechanism of nitrapyrin in the three kinds of study soils, the amount of nitrapyrin adsorption in the test soils was expressed in formula (2.6) (15, 16 and 17) to study whether surface adsorption or the distribution function was dominant in the adsorption process.

Because the fitting parameter of the Freundlich model in this study was more than 1 and the adsorption of nitrapyrin was nonlinear, it was better to adopt a comprehensive adsorption mechanism to explain the adsorption behaviour of nitrapyrin in the three soils. The variation curves of $Q_T$, $Q_P$ and

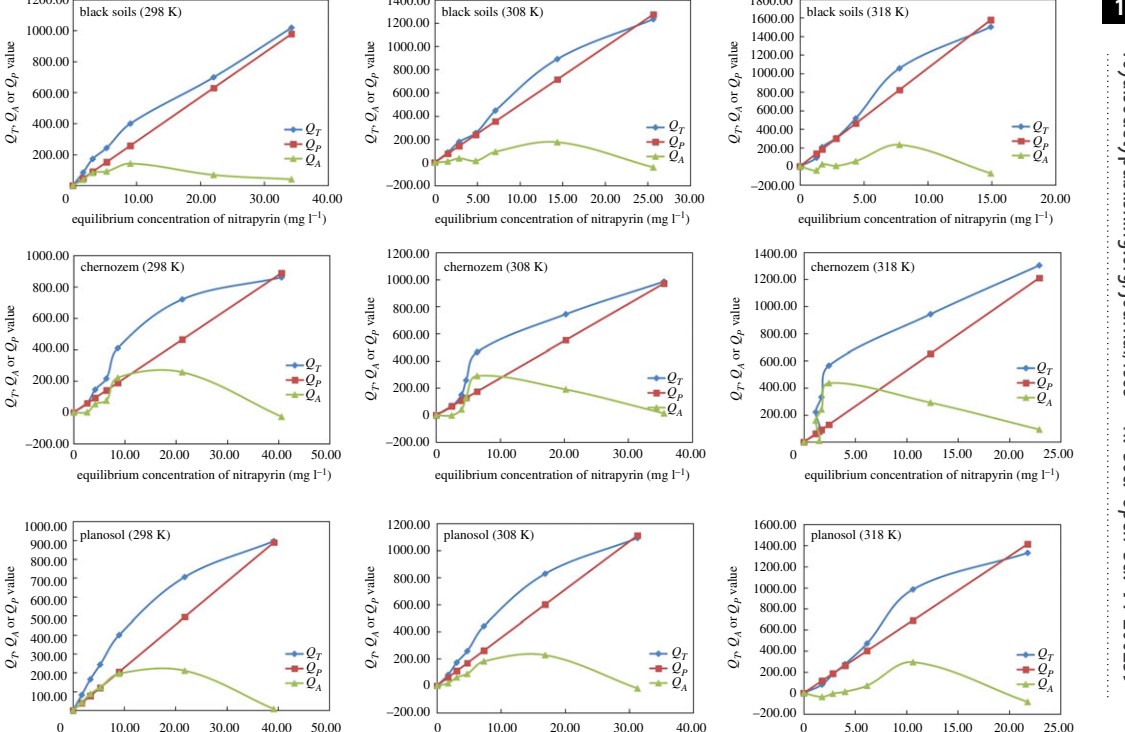

**Figure 3.** The contribution of partitioning and surface adsorption.

$Q_A$ of nitrapyrin with equilibrium solution concentration $C_e$ in the three study soils are shown in figure 3. The adsorption of nitrapyrin in the three soils included two processes: surface adsorption and distribution. However, the contribution of these processes to the adsorption of nitrapyrin was significantly different. Adsorption in the black soil and planosol was mainly due to distribution under different solution concentration conditions; when the solution concentration was low, the adsorption in the chernozem was dominated by surface adsorption. With the increase in the equilibrium solution concentration, the surface adsorption reached saturation and distribution started to play a leading role. Because organic matter in the black soil is higher than other soil, more nitrapyrin was adsorbed by organic matter in black soil. While organic matter in planosol and chernozem was lower, and nitrapyrin was mainly adsorbed by clayey minerals as they were exposed on soil particles' surfaces. Thus, the nitrapyrin adsorption in these three soils was different. With increasing temperature, the contribution of surface adsorption to soil adsorption of nitrapyrin increased.

## 3.6. Effects of different influencing factors

### 3.6.1. Organic matter content

The organic matter applied to soil can be adsorbed by soil organic matter, soil colloids and soil mineral components to different degrees. Generally, chemical adsorption plays an important role in soil with a high content of organic matter. When the content of organic matter was low, physical factors such as soil mineral components played a dominant role. However, compared with soil organic matter, the adsorption caused by soil mineral components was less effective. In this paper, the effect of different soil organic matter contents on the nitrapyrin adsorption capacity was analysed in black soil. After the removal of organic matter by hydrogen peroxide oxidation, the content of organic matter in black soil decreased from $48.51\ \mathrm{g\ kg^{-1}}$ to $44.30\ \mathrm{g\ kg^{-1}}$, $34.37\ \mathrm{g\ kg^{-1}}$, $22.35\ \mathrm{g\ kg^{-1}}$ and $7.27\ \mathrm{g\ kg^{-1}}$, respectively. The adsorption characteristics of black soil before and after removal of the soil organic matter were fitted by the Langmuir equation (table 7).

The maximum buffer capacity (MBC = $Q_m \times k_L$) of soil to nitrapyrin was calculated according to the saturated adsorption capacity $Q_m$ and coefficient $k_L$ calculated by the Langmuir equation [45]. The MBC of organic matter in black soil decreased by 29.79%, 26.38%, 24.99% and 25.10%, respectively.

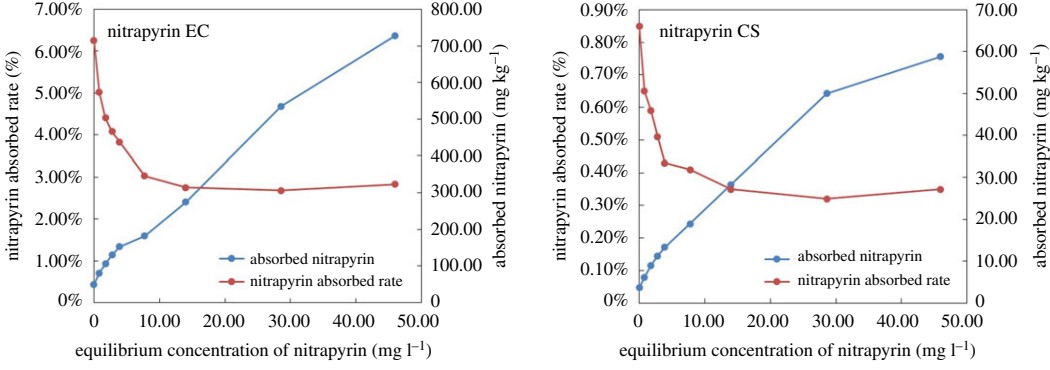

**Figure 4.** The adsorption of different dosage forms of nitrapyrin in black soil.

**Table 7.** Fitting parameters of the adsorption isotherm equation for black soil (298 K).

| treatment | organic matter content (g kg$^{-1}$) | Langmuir model | | |
|---|---|---|---|---|
| | | $q_m$ (mg g$^{-1}$) | $k_L$ (l mg$^{-1}$) | $R^2$ |
| treatment 1 | 48.51 | 2372.3 | 4847.22 | 0.9931 |
| treatment 2 | 44.30 | 1905.1 | 4174.62 | 0.9910 |
| treatment 3 | 34.37 | 1868.4 | 4530.74 | 0.9900 |
| treatment 4 | 22.35 | 1798.3 | 4796.65 | 0.9990 |
| treatment 5 | 7.27 | 1873.7 | 4596.65 | 0.9910 |

With the decrease in soil organic matter content, the buffering capacity of nitrapyrin decreased significantly, indicating that soil organic matter was an important absorbent for nitrapyrin.

The saturated adsorption capacity decreased by 19.69, 21.24, 24.19 and 21.02%. The decrease in organic matter content leads to a decrease in adsorption capacity, which further verifies that organic matter was an important absorbent in soil. However, the MBC and saturated adsorption capacity of treatment 5 were higher than those of treatment 4, indicating that after the content of organic matter in the soil decreased significantly, the adsorption of nitrapyrin by minerals played a certain substituting role. In the process of removing organic matter with hydrogen peroxide, the soil particles decreased and the specific surface area increased, which resulted in more adsorption sites being exposed and the adsorption capacity being promoted.

### 3.6.2. Dosage forms of nitrapyrin

Two dosage forms of nitrapyrin (EC and CS) were selected to analyse its adsorption in black soil at 298 K. The results show that the adsorption capacity and adsorption rate of nitrapyrin EC were significantly higher than those of nitrapyrin CS at different concentrations. The adsorption rate of nitrapyrin EC in black soil was more than seven times higher than that of nitrapyrin CS, with an average value of 7.79%. This shows that different dosage forms could affect the adsorption of nitrapyrin in soil. Compared with traditional dosage forms, nitrapyrin CS showed a significantly reduced adsorption capacity and adsorption rate because nitrapyrin CS was sealed in microcapsules and slowly released, which kept a large amount of nitrapyrin from being adsorbed to the soil. At the same time, the dosage form also provides a new solution to reduce the adsorption of organic matter to nitrapyrin and increase its efficient utilization in high organic matter soil (figure 4).

## 4. Conclusion

The quasi-second-order kinetic equation can be used to fit the adsorption kinetics of nitrapyrin in three kinds of soils ($R^2 \geq 0.8907$, $p < 0.05$). The Langmuir equation can be used to fit the adsorption isotherms of nitrapyrin in three kinds of soils ($R^2 \geq 0.9400$, $p < 0.05$), and the maximum adsorption capacity at

different temperatures were black soil > planosol > chernozem. The black soil and planosol mainly achieve adsorption by distribution under different solution concentration conditions. The adsorption of chernozem included the surface adsorption when the solution concentration was low and the distribution process when the equilibrium solution concentration was high. The organic matter and dosage form can significantly affect the adsorption of nitrapyrin on soils. The decrease in organic matter concentration led to the decrease in adsorption capacity. The adsorption rate of black soil to nitrapyrin EC was much higher than that of nitrapyrin CS.

Data accessibility. It is provided as electronic supplementary material.

Authors' contributions. Z.Z. carried out the laboratory work, participated in data analysis, participated in the design of the study and drafted the manuscript; J.Y., Y.L., J.L. and H.S. participated in data analysis; Yu.W., Yi.W., S.W. and G.F. participated in the design of the study and drafted the manuscript; Q.G. conceived of the study, designed the study, coordinated the study and helped draft the manuscript. All authors gave final approval for publication and agree to be held accountable for the work performed therein.

Competing interests. We declare we have no competing interests.

Funding. The present study was supported by grants from the National Key Research and Development Program of China (grant nos 2016YFD0200403, 2016YFD0200101) and the Scientific Technology Development Research Plan Project of Jilin Province in China (grant no. 20180201035NY).

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
