## [Reviewer comments · Royal Society Open Science]

Review History

RSOS-191684.R0 (Original submission)

Review form: Reviewer 1

Is the manuscript scientifically sound in its present form?

Yes

Are the interpretations and conclusions justified by the results?

Yes

Is the language acceptable?

Yes

Do you have any ethical concerns with this paper?

No

Have you any concerns about statistical analyses in this paper?

No

Recommendation?

Major revision is needed (please make suggestions in comments)

Comments to the Author(s)

This work assesses the adsorption mechanism of nitrapyrin in three types of soils (black soil, chernozem and planosol) in northeast China. It discusses the adsorption kinetic and adsorption thermodynamics of nitrapyrin in the selected soil, and the adsorption rate of black soil to different dosage forms of nitrapyrin under different organic matter content. This study is meaningful for the adsorption of nitrapyrin in the soil environment. However, some revisions should be made to make the paper more convincing.

1. There are some grammatical errors in the manuscript. For example, Line 369, "study soils are shown in Figure 3" should be placed with "study soils were shown in Figure 3". And Line 326, "are listed in Table 6." should be "were listed in Table 6." etc.
2. Why do you choose the two dosage forms of nitrapyrin?
3. Line 334, what might be the chemical reaction? Why do not chemical adsorption occur in black soil?
4. Check the Fig. 2 and 3. For example, the curves are different but the names are the same in Fig. 2.
5. Line 370, why the contribution of the two processes to the adsorption of nitrapyrin was significantly different in three types of soils?
6. The conclusions are too simple to highlight the more profound significance of the study.
7. Lack error bars.

Review form: Reviewer 2

Is the manuscript scientifically sound in its present form?

No

Are the interpretations and conclusions justified by the results?

Yes

Is the language acceptable?

Yes

Do you have any ethical concerns with this paper?

No

Have you any concerns about statistical analyses in this paper?

No

Recommendation?

Reject

Comments to the Author(s)

Zhang and Gao conducted a study to assess the adsorption of the nitrification inhibitor nitrapyrin in different types of soils. This manuscript does not present novel results. The outcomes of this research are already expected. Jacinthe and Pichtel 1992, discussed the mechanism and factor affecting adsorption of Nitrapyrin in various soils in details.

Decision letter (RSOS-191684.R0)

30-Jan-2020

Dear Professor Gao:

Manuscript ID: RSOS-191684

Title: "The adsorption and mechanism of the nitrification inhibitor nitrapyrin in different types of soils"

Thank you for submitting the above manuscript to Royal Society Open Science. Your paper was sent to reviewers and their comments are included at the bottom of this letter. I apologise this has taken longer than usual.

In view of the concerns raised by the reviewers, the manuscript has been rejected in its current form. However, a new manuscript may be submitted which takes into consideration these comments.

Please note that resubmitting your manuscript does not guarantee eventual acceptance, and that your resubmission will be subject to peer review before a decision is made.

Your resubmitted manuscript should be submitted by 29-Jul-2020. If you are unable to submit by this date please contact the Editorial Office.

On behalf of the Subject Editor Professor Anthony Stace and the Associate Editor Professor Tobias Hertel

REVIEWER(S) REPORTS:
Associate Editor Comments to Author ():
RSC Associate Editor:
Comments to the Author:
30-Jan-2020

RSOS-191684 - The adsorption and mechanism of the nitrification inhibitor nitrapyrin in different types of soils

Dear Professor Gao,

we regret to inform you that your manuscript has been rejected by our reviewers. We will, however, gladly review a resubmitted and thoroughly revised version of this manuscript that addresses the concerns voiced by our reviewers.

Most significantly the criticism was directed at a perceived lack of novelty since Jacinthe and Pichtel 1992, also cited in your manuscript, were mentioned as having previously discussed the mechanism and factors affecting adsorption of Nitrapyrin in various soils in detail. Other points of criticism were directed at the lack of error bars and margins which allow readers to assess the significance of your data.

Should you decide to resubmit a thoroughly revised version of this manuscript it should thus perhaps include a clear discussion of how and in what aspects this research goes beyond previous findings and how specifically it expands the current understanding of these reactions and their kinetics.

Kind regards,
Professor Tobias Hertel
Royal Society Open Science
openscience@royalsociety.org

RSC Subject Editor:
Comments to the Author:
(There are no comments.)

Reviewers' Comments to Author:
Reviewer: 1

Comments to the Author(s)

This work assesses the adsorption mechanism of nitrapyrin in three types of soils (black soil, chernozem and planosol) in northeast China. It discusses the adsorption kinetic and adsorption thermodynamics of nitrapyrin in the selected soil, and the adsorption rate of black soil to different dosage forms of nitrapyrin under different organic matter content. This study is meaningful for the adsorption of nitrapyrin in the soil environment. However, some revisions should be made to make the paper more convincing.

1. There are some grammatical errors in the manuscript. For example, Line 369, "study soils are shown in Figure 3" should be placed with "study soils were shown in Figure 3". And Line 326, "are listed in Table 6." should be "were listed in Table 6." etc.
2. Why do you choose the two dosage forms of nitrapyrin?
3. Line 334, what might be the chemical reaction? Why do not chemical adsorption occur in black soil?
4. Check the Fig. 2 and 3. For example, the curves are different but the names are the same in Fig. 2.
5. Line 370, why the contribution of the two processes to the adsorption of nitrapyrin was significantly different in three types of soils?
6. The conclusions are too simple to highlight the more profound significance of the study.
7. Lack error bars.

Reviewer: 2

Comments to the Author(s)

Zhang and Gao conducted a study to assess the adsorption of the nitrification inhibitor nitrapyrin in different types of soils. This manuscript does not present novel results. The outcomes of this research are already expected. Jacinthe and Pichtel 1992, discussed the mechanism and factor affecting adsorption of Nitrapyrin in various soils in details.

Author's Response to Decision Letter for (RSOS-191684.R0)

See Appendix A.

RSOS-200259.R0

Review form: Reviewer 1

Is the manuscript scientifically sound in its present form?

Yes

Are the interpretations and conclusions justified by the results?

Yes

Is the language acceptable?

Yes

Do you have any ethical concerns with this paper?

No

Have you any concerns about statistical analyses in this paper?

No

Recommendation?

Accept with minor revision (please list in comments)

Comments to the Author(s)

Conditions, this paper studied the adsorption kinetics and isotherm adsorption of nitrapyrin in three types of soils. The factors affecting the adsorption capacity and the potential adsorption mechanisms of different soils are analyzed. The significance of this work is based on a practical problem, an ongoing Chinese policy, which provides the work with practical value. Therefore, I recommend a minor revision before its publication. The specific comments are listed below:

1. The abstract section lacked a brief summary of the background and significance of the research.
2. Please add the reference of the potassium dichromate volumetric method.
3. The original content of the organic matter was different in the three types of soil, does all samples conduct the H₂O₂ treatment in the same way?
4. Please modified the format of the table 3.

Review form: Reviewer 3

Is the manuscript scientifically sound in its present form?

Yes

Are the interpretations and conclusions justified by the results?

Yes

Is the language acceptable?

No

Do you have any ethical concerns with this paper?

No

Have you any concerns about statistical analyses in this paper?

No

Recommendation?

Major revision is needed (please make suggestions in comments)

Comments to the Author(s)

1. Line 37, "caused by excessive nitrogen fertilizer use" should be revised as "caused by the excessive use of nitrogen fertilizer".
 2. Line 40, "Goring et al. (Goring et al., 1962) first reported that" should be revised as "Goring et al. (1962) first reported that"
 3. Line 82, when "NUE" first appears in the full text, please use its full name.
 4. Line 110, "For organic matter removal from the soil" should be revised as "For removal of organic matter from the soil".
 5. Line 138, when "HPLC" first appears in the full text, please use its full name: high performance liquid chromatography.
 6. Line 247, "which", "and" are not used correctly.
 7. The format of Table 3 is not well adjusted.
 8. Line 279, "because $\Delta G^\#$ was more than 0" should be modified as "because $\Delta G^\#$ was larger than 0".
 9. Line 335, "chernozem at 318K and planosol at 308 K and 318 K" should be modified as "chernozem(318 K) and planosol (308 K and 318 K)".
 9. Line 399, "29.79, 26.38, 24.99, and 25.10%" should be modified as "29.79%, 26.38%, 24.99% and 25.10%".
 10. Line 402, "19.69, 21.24, 24.19 and 21.02%" should be modified as "19.69%, 21.24%, 24.19% and 21.02%".
 11. The past tense should be revised in the entire manuscript.
 12. Line 434, "The decrease in organic matter content leads to the" should be modified as "The decrease in organic matter concentration led to the"
 13. Line 436, "nitrapyrin EC is more higher than that of nitrapyrin CS" should be modified as "nitrapyrin EC is much higher than that of nitrapyrin CS"
 14. The similar mistakes mentioned above should be revised in the entire manuscript carefully.
 15. The mechanism of adsorption of nitrapyrin in different types of soils should be concluded and analyzed clearly.
 16. Why only do the impact of organic matter content and nitrapyrin concentration on the adsorption of black soil? Previous studies have already investigated the impact of organic matter content and nitrapyrin concentration on the adsorption of black soil. Please explain the novelty of this research by comparing the previous reports.
- Effect of Soil Organic Matter on Adsorption of Nitrification Inhibitor Nitrapyrin in Black Soil:
COMMUNICATIONS IN SOIL SCIENCE AND PLANT ANALYSIS, 51(7):883-895.

Decision letter (RSOS-200259.R0)

Dear Professor Gao:

Title: The adsorption and mechanism of the nitrification inhibitor nitrapyrin in different types of soils

Manuscript ID: RSOS-200259

Thank you for submitting the above manuscript to Royal Society Open Science. On behalf of the Editors and the Royal Society of Chemistry, I am pleased to inform you that your manuscript will be accepted for publication in Royal Society Open Science subject to minor revision in accordance with the referee suggestions. Please find the reviewers' comments at the end of this email.

The reviewers and handling editors have recommended publication, but also suggest some minor revisions to your manuscript. Therefore, I invite you to respond to the comments and revise your manuscript.

Because the schedule for publication is very tight, it is a condition of publication that you submit the revised version of your manuscript before 19-Jul-2020. Please note that the revision deadline will expire at 00.00am on this date. If you do not think you will be able to meet this date please let me know immediately.

Kind regards,
Dr Laura Smith
Publishing Editor, Journals

On behalf of the Subject Editor Professor Anthony Stace and the Associate Editor Professor Tobias Hertel.

RSC Associate Editor
Comments to the Author:
Dear Dr. Gao,

we hope you found the reviewers comments helpful when revising your manuscript.

We would like to publish your manuscript with minor revisions. Specifically, we noticed that some of the comments of reviewer 2 have been left unanswered and that no changes were made to the manuscript regarding several of his/her points of criticism.

Before proceeding with publication of this manuscript we would thus like to ask you to correct the following items including the last two which were added by the associate editor:

- add the unabbreviated form of NUE in line 82
- table 3 is difficult to read, we ask you to revise the format to make it more accessible
- with any reported quantity, use significant digits only. Suppose the error margins for Delta G values in table 4, for example, were 0.8 kJ/mol (here I am just guessing), this would suggest that the second numeral would be the significant digit. In this example 85 kJ/mol would be the appropriate quantity to report instead of 85.213 kJ/mol.
- the use of a hashtag with thermodynamic potentials is highly unusual, use a double dagger instead whenever potentials are referenced to a transition state.

Sincerely,
Tobias Hertel
Associate Editor, RSOS

Reviewer comments to Author:

Reviewer: 1

Comments to the Author(s)

conditions, this paper studied the adsorption kinetics and isotherm adsorption of nitrapyrin in three types of soils. The factors affecting the adsorption capacity and the potential adsorption mechanisms of different soils are analyzed. The significance of this work is based on a practical problem, an ongoing Chinese policy, which provides the work with practical value. Therefore, I recommend a minor revision before its publication. The specific comments are listed below:

1. The abstract section lacked a brief summary of the background and significance of the research.
2. Please add the reference of the potassium dichromate volumetric method.
3. The original content of the organic matter was different in the three types of soil, does all samples conduct the H₂O₂ treatment in the same way?
4. Please modified the format of the table 3.

Reviewer: 3

Comments to the Author(s)

1. Line 37, "caused by excessive nitrogen fertilizer use" should be revised as "caused by the excessive use of nitrogen fertilizer".
2. Line 40, "Goring et al. (Goring et al., 1962) first reported that" should be revised as "Goring et al. (1962) first reported that"
3. Line 82, when "NUE" first appears in the full text, please use its full name.
4. Line 110, "For organic matter removal from the soil" should be revised as "For removal of organic matter from the soil".
5. Line 138, when "HPLC" first appears in the full text, please use its full name: high performance liquid chromatography.
6. Line 247, "which", "and" are not used correctly.
7. The format of Table 3 is not well adjusted.
8. Line 279, "because $\Delta G^\#$ was more than 0" should be modified as "because $\Delta G^\#$ was larger than 0".
9. Line 335, "chernozem at 318K and planosol at 308 K and 318 K" should be modified as "chernozem(318 K) and planosol (308 K and 318 K).
9. Line 399, "29.79, 26.38, 24.99, and 25.10%" should be modified as "29.79%, 26.38%, 24.99% and 25.10%".
10. Line 402, "19.69, 21.24, 24.19 and 21.02%" should be modified as "19.69%, 21.24%, 24.19% and 21.02%".
11. The past tense should be revised in the entire manuscript.
12. Line 434, "The decrease in organic matter content leads to the" should be modified as "The decrease in organic matter concentration led to the"
13. Line 436, "nitrapyrin EC is more higher than that of nitrapyrin CS" should be modified as "nitrapyrin EC is much higher than that of nitrapyrin CS"
14. The similar mistakes mentioned above should be revised in the entire manuscript carefully.
15. The mechanism of adsorption of nitrapyrin in different types of soils should be concluded and analyzed clearly.
16. Why only do the impact of organic matter content and nitrapyrin concentration on the adsorption of black soil? Previous studies have already investigated the impact of organic matter content and nitrapyrin concentration on the adsorption of black soil. Please explain the novelty of this research by comparing the previous reports.

Effect of Soil Organic Matter on Adsorption of Nitrification Inhibitor Nitrapyrin in Black Soil: COMMUNICATIONS IN SOIL SCIENCE AND PLANT ANALYSIS, 51(7):883-895.

Author's Response to Decision Letter for (RSOS-200259.R0)

See Appendix B.

Decision letter (RSOS-200259.R1)

Dear Professor Gao:

Title: The adsorption and mechanism of the nitrification inhibitor nitrapyrin in different types of soils

Manuscript ID: RSOS-200259.R1

It is a pleasure to accept your manuscript in its current form for publication in Royal Society Open Science. The chemistry content of Royal Society Open Science is published in collaboration with the Royal Society of Chemistry.

On behalf of the Subject Editor Professor Anthony Stace and the Associate Editor Professor Tobias Hertel.

RSC Associate Editor
Comments to the Author:
(There are no comments.)

Reviewer(s)' Comments to Author:

Appendix A

Response to Reviewers

Dear Reviewer 1:

Thank you for taking so much time on reviewing my manuscript and giving me so many sincere comments. I have carefully revised all your suggestions as follows. Substantia revision were made (**Please reference to "Red marked in the revised manuscript"**).

The problem: (1) There are some grammatical errors in the manuscript. For example, Line 369, “study soils are shown in Figure 3” should be placed with “study soils were shown in Figure 3” . And Line 326, “are listed in Table 6.” should be “were listed in Table 6.” etc.

Answer: Thanks for your good advice, the grammatical errors in the manuscript have been seriously modified.

The problem: (2) Why do you choose the two dosage forms of nitrapyrin?

Answer: Previous studies have shown that nitrapyrin was easily adsorbed by soil organic matter when it was applied to soil (Huang et al., 2001), especially for soils with high level of organic matter, and its availability was significantly reduced (Hendrickson et al., 1979; Chen et al., 1980; Sahrawat et al., 1987). So, in this study we choose two dosage forms of nitrapyrin and try to find which dosage form is better to reduce the adsorption of organic matter in the soils and improve the effect of

nitrapyrin. The results provide basic theoretical support for developing **new dosage forms of nitrapyrin (Fabrication and release behavior of nitrapyrin Microcapsules: Using modified melamine-formaldehyde resin as shell material, Science of the Total Environment, 704 (2020) 135394).**

The problem: (3) Line 334, what might be the chemical reaction? Why do not chemical adsorption occur in black soil?

Answer: The organic matter applied to soil can be adsorbed by soil organic matter, soil colloids and mineral components in different degrees. Generally, as organic matter in black soil is high, the nitrapyrin were largely adsorbed. The increase in temperature was little beneficial to the process of chemical adsorption of nitrapyrin in black soil. So, in this paper, the effect of different soil organic matter contents on the nitrapyrin adsorption capacity was mainly analyzed in black soil.

The problem: (4) Check the Fig. 2 and 3. For example, the curves are different but the names are the same in Fig. 2.

Answer: Thanks again for your good advice, the Fig. 2 and 3 were checked seriously.

The problem: (5) Line 370, why the contribution of the two processes to the adsorption of nitrapyrin was significantly different in three types of soils?

Answer: The effect of organic matter content on the nitrapyrin adsorption capacity was mainly analyzed in black soil. The clay and mineral structure of soil may be the main reason of the chernozem and planosol (This part will focus on study in the next step) that the contribution of the two processes to the adsorption of nitrapyrin was significantly different.

The problem: (6) The conclusions are too simple to highlight the more profound significance of the study.

Answer: Thanks again for your good advice, the conclusion has been seriously revised (line 426-437).

The problem: (7) Lack error bars.

Answer: The error bars were added on figure1. I think other figures are not necessary to add the error bars, because they are fitted curves and there is no significant analysis between different treatments. **At the same time, I also refer to similar articles in recent years (Liu et al., 2017; Yan et al., 2018. etc.).**

Dear Reviewer 2:

Thank you for taking time out of your busy schedule to review my manuscript and give me the sincere comment.

The problem: This manuscript does not present novel results. The outcomes of this research are already expected. Jacinthe and Pichtel 1992,

discussed the mechanism and factor affecting adsorption of Nitrapyrin in various soils in details.

Answer: Jacinthe and Pichtel (1992) only discussed the adsorption mechanism of nitrapyrin on humic acid and fulvic acid, which are the main components of soil organic matter. The adsorption mechanism of nitrapyrin might be different in different soil types with different clay and mineral structure characteristics compared with humic acid and fulvic acid of soil organic matter. So, The present work is an attempt to assess the adsorption mechanism on adsorption behavior of nitrapyrin on three soils in china, which will help to determine the dose of nitrapyrin required for availability of this nitrification inhibitor in different types of soils and to provide a theoretical basis for elucidating the adsorption of nitrapyrin in the soil environment. The importance of this study is as follows: (1) The contribution of surface adsorption and distribution of three types of soils to the adsorption of nitrapyrin was discussed. (2) The effect of different soil organic matter contents on the nitrapyrin adsorption capacity was analyzed in black soil. After the content of organic matter in the soil being decreased significantly, the adsorption of nitrapyrin by minerals played a certain substituting role. (3) we choose two dosage forms of nitrapyrin and try to find which dosage form can reduce the adsorption of organic matter in the soils and improve the effect of nitrapyrin application. The results provide a basic theoretical support

for developing **new dosage forms of nitrapyrin** (Fabrication and release behavior of nitrapyrin Microcapsules: Using modified melamine-formaldehyde resin as shell material, Science of the Total Environment, 704 (2020) 135394).

We are looking forward to hearing from you soon. If you have any question, please connect me without hesitation. Thank you sincerely.

Yours Sincerely, Zhong-qing Zhang.

Appendix B

Response to Editor

Dear Editor:

Thank you for taking time out of your busy schedule to review my manuscript and give me so many sincere comments. I have carefully examined the point provided. (Please reference to "Blue marked in the revised manuscript").

The problem: (1) add the unabbreviated form of NUE in line 82.

Answer: Thanks for your good advice, the unabbreviated form of NUE (nitrogen use efficiency) was added in line 87.

The problem: (2) table 3 is difficult to read, we ask you to revise the format to make it more accessible.

Answer: The format of the table 3 have been revised in the manuscript.

The problem: (3) with any reported quantity, use significant digits only. Suppose the error margins for Delta G values in table 4, for example, were 0.8 kJ/mol (here I am just guessing), this would suggest that the second numeral would be the significant digit. In this example 85 kJ/mol would be the appropriate quantity to report instead of 85.213 kJ/mol.

Answer: It has been carefully revised. (table4 and table6)

The problem: (4) the use of a hashtag with thermodynamic potentials is highly unusual, use a double dagger instead whenever potentials are referenced to a transition state.

Answer: Thanks for your good advice, we have reference the literature of Chowdhury et al.(2011), Won et al.(2006) and Yan et al.(2018)that before the use of a hashtag with thermodynamic potentials.

Chowdhury S, Mishra R, Saha P, et al.2011.Adsorption thermodynamics, kinetics and isosteric heat of adsorption of malachite green onto chemically modified rice husk [J].Desalination, 265(1-3): 159-168.

Won S W, Kim H-J, Choi S-H, et al.2006. Performance, kinetics and equilibrium in biosorption of anionic dye Reactive Black 5 by the waste biomass of Corynebacterium glutamicum as a low-cost biosorbent [J].Chemical Engineering Journal, 121 (1):37-43.

YAN Aichun, XIE Xiuhong, FAN Chunying, ZHANG Zhidan, ZHENG Lirong, ZHANG Jinjing. Adsorption behavior and mechanism of Cu (II) on soil humin [J]. Acta Scientiae Circumstantiae, 2018, 38(12):4779-4788.

Response to Reviewers

Dear Reviewer 1:

Thank you for taking so much time on reviewing my manuscript and giving me so many sincere comments. I have carefully revised all your suggestions as follows. Substantia revision were made (**Please reference to "Red marked in the revised manuscript"**).

The problem: (1) The abstract section lacked a brief summary of the background and significance of the research.

Answer: The abstract background and significance of the research has been added. (line23-25, 36-38)

The problem: (2) Please add the reference of the potassium dichromate volumetric method.

Answer: The reference of the potassium dichromate volumetric method was added in the research. Reference the book "**Technical guidance for soil agrochemical analysis**".

The problem: (3) The original content of the organic matter was different in the three types of soil, does all samples conduct the H₂O₂ treatment in the same way?

Answer: The original content of the organic matter was different in the three types of soil, but only the black soil conduct the H₂O₂ treatment in this study.

The problem: (4) Please modified the format of the table 3.

Answer: The format of the table 3 have been revised in the manuscript.

Dear Reviewer 2:

Thank you for taking time out of your busy schedule to review my manuscript and give me the sincere comment (**Please reference to "Red marked in the revised manuscript"**).

The problem: (1) Line 37, "caused by excessive nitrogen fertilizer use" should be revised as "caused by the excessive use of nitrogen fertilizer".

Answer: It has been modified in manuscript with the requirements.

The problem: (2) Line 40, "Goring et al. (Goring et al., 1962) first reported that" should be revised as " Goring et al. (1962) first reported that".

Answer: It has been modified in manuscript with the requirements.

The problem: (3) Line 82, when "NUE" first appears in the full text, please use its full name.

Answer: It has been modified in manuscript with the requirements.

The problem: (4) Line 110, "For organic matter removal from the soil" should be revised as "For removal of organic matter from the soil".

Answer: It has been modified in manuscript with the requirements.

The problem: (5) Line 138, when "HPLC" first appears in the full text, please use its full name: high performance liquid chromatography.

Answer: It has been modified in manuscript with the requirements.

The problem: (6) Line 247, "which", "and" are not used correctly.

Answer: It has been modified in manuscript with the requirements. (line 255-257, **Figure 1 shows the adsorption kinetics curves of nitrapyrin in three soil types with initially rapid adsorption stage and following slow adsorption stage to the equilibrium after 400 min.**)

The problem: (7) The format of Table 3 is not well adjusted.

Answer: The format of the table 3 have been revised in the manuscript.

The problem: (8) Line 279, "because ΔG^\ddagger was more than 0" should be modified as "because ΔG^\ddagger was larger than 0".

Answer: It has been modified in manuscript with the requirements.

The problem: (9) Line 335, "chernozem at 318K and planosol at 308 K and 318 K" should be modified as " chernozem(318 K) and planosol (308 K and 318 K).

Answer: It has been modified in manuscript with the requirements.

The problem: (10) Line 399, "29.79, 26.38, 24.99, and 25.10%" should be modified as "29.79%, 26.38%, 24.99% and 25.10%".

Answer: It has been modified in manuscript with the requirements.

The problem: (11) Line 402, "19.69, 21.24, 24.19 and 21.02%" should be modified as "19.69%, 21.24%, 24.19% and 21.02%".

Answer: It has been modified in manuscript with the requirements.

The problem: (12) The past tense should be revised in the entire manuscript.

Answer: It has been modified in manuscript with the requirements.

The problem: (13) Line 434, "The decrease in organic matter content leads to the" should be modified as "The decrease in organic matter concentration led to the"

Answer: It has been modified in manuscript with the requirements.

The problem: (14) Line 436, "nitrapyrin EC is more higher than that of nitrapyrin CS" should be modified as "nitrapyrin EC is much higher than that of nitrapyrin CS"

Answer: It has been modified in manuscript with the requirements.

The problem: (15) The similar mistakes mentioned above should be revised in the entire manuscript carefully.

Answer: It has been modified in manuscript with the requirements.

The problem: (16) The mechanism of adsorption of nitrapyrin in different types of soils should be concluded and analyzed clearly.

Answer: Thank you very much for your valuable advice. The adsorption mechanism of nitrapyrin on different types of soils were further concluded and analyzed. (line 386-390)

The problem: (17) Why only do the impact of organic matter content and nitrapyrin concentration on the adsorption of black soil? Previous studies have already investigated the impact of organic matter content and nitrapyrin concentration on the adsorption of black soil. Please explain the novelty of this research by comparing the previous reports.

Answer: **The nitrapyrin was easily adsorbed by soil, but most current studies are focused on comparing the effects of nitrapyrin application at different soil organic matter levels and in different soil types.** The adsorption kinetics and isotherm adsorption of the nitrification inhibitor nitrapyrin in black soil, chernozem and planosol were studied in this paper. The present work is an attempt to assess the adsorption mechanism on adsorption behaviour of nitrapyrin on three soils of china, which will help to determine the dose of nitrapyrin required for availability of this nitrification inhibitor in different types of soils and to provide a theoretical basis for elucidating the adsorption of nitrapyrin in the soil environment.

The study found that the adsorption capacity and rate of nitrapyrin in the soil decreased with decreasing soil organic matter. Therefore, an attempt was made to preliminarily explore the influence of organic matter on adsorption of nitrapyrin.

The study of “Effect of Soil Organic Matter on Adsorption of Nitrification Inhibitor Nitrapyrin in Black Soil” was also our research

and was an important extension of this research. The paper mainly discussed the influence of organic matter on adsorption of nitrapyrin.

At the same time, this study provides basic theoretical support for developing new dosage forms of nitrapyrin to reduce the adsorption of organic matter and improve the effectiveness of nitrapyrin application.

We are looking forward to hearing from you soon. If you have any question, please connect me without hesitation. Thank you sincerely.

Yours Sincerely, Zhong-qing Zhang.